# Quantitative Trait Loci for Genotype and Genotype by Environment Interaction Effects for Seed Yield Plasticity to Terminal Water-Deficit Conditions in Canola (*Brassica napus* L.)

**DOI:** 10.3390/plants12040720

**Published:** 2023-02-06

**Authors:** Harsh Raman, Nawar Shamaya, Ramethaa Pirathiban, Brett McVittie, Rosy Raman, Brian Cullis, Andrew Easton

**Affiliations:** 1NSW Department of Primary Industries, Wagga Wagga Agricultural Institute, Wagga Wagga, NSW 2650, Australia; 2Centre for Biometrics and Data Science for Sustainable Primary Industries, National Institute for Applied Statistics Research Australia, University of Wollongong, Wollongong, NSW 2522, Australia; 3Advanta Seeds Pty Ltd., 268 Anzac Avenue, Toowoomba, QLD 4350, Australia

**Keywords:** canola, quantitative trait loci (QTL), water use efficiency, drought tolerance, genotype by environment interaction

## Abstract

Canola plants suffer severe crop yield and oil content reductions when exposed to water-deficit conditions, especially during the reproductive stages of plant development. There is a pressing need to develop canola cultivars that can perform better under increased water-deficit conditions with changing weather patterns. In this study, we analysed genetic determinants for the main effects of quantitative trait loci (QTL), (Q), and the interaction effects of QTL and Environment (QE) underlying seed yield and related traits utilising 223 doubled haploid (DH) lines of canola in well-watered and water-deficit conditions under a rainout shelter. Moderate water-deficit at the pre-flowering stage reduced the seed yield to 40.8%. Multi-environmental QTL analysis revealed 23 genomic regions associated with days to flower (DTF), plant height (PH) and seed yield (SY) under well-watered and water-deficit conditions. Three seed yield QTL for main effects were identified on chromosomes A09, C03, and C09, while two were related to QE interactions on A02 and C09. Two QTL regions were co-localised to similar genomic regions for flowering time and seed yield (A09) and the second for plant height and chlorophyll content. The A09 QTL was co-located with a previously mapped QTL for carbon isotope discrimination (Δ^13^C) that showed a positive relationship with seed yield in the same population. Opposite allelic effects for plasticity in seed yield were identified due to QE interactions in response to water stress on chromosomes A02 and C09. Our results showed that QTL’s allelic effects for DTF, PH, and SY and their correlation with Δ^13^C are stable across environments (field conditions, previous study) and contrasting water regimes (this study). The QTL and DH lines that showed high yield under well-watered and water-deficit conditions could be used to manipulate water-use efficiency for breeding improved canola cultivars.

## 1. Introduction

Drought (water-deficit) is a major abiotic stress that adversely affects crop production worldwide. In the past decade, global losses in crop production due to drought have been estimated at approximately $30 billion [1]. Climate models predict drought will be the new norm and affect crop productivity globally with increasing global temperatures, meteorological drought, and reduced water availability for agriculture. Plants lose 100 to 400 water molecules for transpiration for every carbon atom fixed by photosynthesis [2]. This trade-off highlights the pressing need to identify cultivars that can perform better under increased drought conditions. 

Canola (*Brassica napus* L.) is grown worldwide to mainly produce healthy vegetable oil, biodiesel, and protein-rich feedstock. It is grown as a broadacre crop in several arid and semi-arid regions, where its production relies heavily on stored soil moisture or in-season rainfall. Canola often faces periodic drought events, especially at the reproductive stages of plant development, that cause severe crop yield and oil content reduction [1,2,3]. A considerable genetic variation in traits implicated in drought escape and drought avoidance strategies has been identified in several crops, including canola [4,5,6,7,8,9]. Studies have shown that early flowering is negatively associated with seed yield in a large selection of canola lines under natural field conditions [10,11]. However, the drought escape strategy often compromises the crop yield potential under non-water-limited environments [12]. Drought avoidance traits such as leaf carbon isotope discrimination (Δ^13^C) and root traits (root pulling force, root length and root biomass) enable plants to maintain water status at a high vapour pressure deficit and were associated with improved water use efficiency (WUE) and seed yield in canola. Many quantitative trait loci (QTL) underlying variation for drought escape traits, seed yield and drought avoidance traits have been detected in canola [13,14,15,16]. However, few studies have reported QTL and correlated phenotypes for response to water stress under contrasting rainfed (DRY) and irrigated (WET) environments in canola [11,13,14].

Yield response to water-deficit conditions is difficult to assess under field conditions. The significance of the treatment effect depends upon natural precipitation, and the impact of many abiotic and biotic factors converges and complicates the identification of component traits involved in drought resistance. In addition to leaf Δ^13^C and root traits (root pulling force, root length and root biomass), positive correlations between leaf chlorophyll concentration expressed as soil-plant analytical development (SPAD) chlorophyll meter reading (SCMR), transpiration efficiency (TE) and adaptation to drought have been reported in some crop plants [17,18,19]. However, the relationship between simple-to-measure phenotype, SPAD, and seed yield, and relevant loci controlling variation in SPAD have not been reported in canola yet. 

Two canola breeding lines, BC1329 and BC9102, have shown considerable variation in transpiration efficiency (TE). Both parental lines showed a negative relationship with Δ^13^C and *i*WUE, measured as a ratio between light-saturated assimilation rate (*A*) and stomatal conductance to the diffusion of water vapour (*g_sw_*) [20]. In the doubled haploid (DH) population derived from BC1329/BC9102, Raman et al. [15] reported a total of 29 QTL (15 QTL for main effects—corresponding to genotypic (G) effects—and 14 for QTL (Q) by Environment (E) interactions (QE)—corresponding to genotype by environment (GE) effects for variation in drought avoidance traits—Δ^13^C, early vigour, and plant height and drought escape traits such as flowering time in canola under field and shade-house conditions). However, the relevance of those QTL and phenotypic correlations between yield-related traits in contrasting water regimes was not detailed in the complete set of DH lines derived from BC1329 and BC9102. Herein, we aim to investigate the QTL associated with G and GE interaction effects for the plasticity of seed yield and its related traits in response to water stress imposed at the flowering time in a DH population derived from BC13299/BC9102 under semi-controlled rainout shelter conditions. We followed a multi-environment QTL analysis approach to identify QTL associated with main effects and QE interaction, and then compared them with those identified in our earlier study [15]. 

## 2. Materials and Methods

### 2.1. Genetic Material

The study utilised a total of 223 lines from a doubled haploid (DH, designated as 06-5101) population derived from the F_1_ cross between advanced Australian breeding lines ‘BC1329’ (maternal parent) and ‘BC9102’ (paternal parent) along with the parental lines [15] and five check cultivars: Ag-Spectrum, Charlton, Monty, Skipton, and Tarcoola-22. The check cultivars were included to select the candidate DH lines for improved seed yield under water stress relative to benchmark genotypes. The parental lines showed considerable genetic variation for various traits: normalised difference in the vegetative index (NDVI), specific leaf weight, leaf water content, *A*, *g_sw_*, Δ^13^C, days to flower, plant height, seed yield and transpiration efficiency [15,20]. The QTL associated with variation for some of the traits mentioned above were also mapped in our previous study [15].

### 2.2. Experimental Design

An experiment was conducted during the winter growing season (April–December) at the Wagga Wagga Agricultural Institute (latitude: 35°03′01″ S, longitude: 147°21′03″ E), NSW, Australia, in 2018. The experimental site was drill-cultivated, and pre-emergence herbicides (glyphosate) were applied to control summer weeds. A granular fertiliser (N:P:K:S, 22:1:0:15) was applied at 150 kg ha^−1^ in the soil before sowing plots. An array of five portable rainout shelters (ROS, the configuration of each was 15 m wide × 15 m long) were moved to the experiment site. The rainout shelters were designed and manufactured by Southern Central Engineering Pty. Ltd. (Leeton, NSW, Australia) and Polytex (Leeton, NSW, Australia) as per specifications outlined by the National Brassica Germplasm Improvement Program (NSW DPI and GRDC). Details of the ROS are given (Appendix A). All lines were sown, raised, and further evaluated within the ROS (15 m wide × 75 m long) under well-watered and water-deficit conditions. 

Under the ROS, there were two adjacent watering blocks, named well-watered and water-deficit, where both watering blocks were managed with the same protocol except for watering. Plots were arranged in rectangular arrays within each watering block with 20 rows and 23 ranges (Appendix A). The treatments comprised the factorial combinations of water regimes: well-watered (WW, control) and water-deficit (WD, stress); and lines (genotypes). The treatments were allocated to plots in such a way that each watering regime was allocated to all plots in a single watering block (for operational convenience), and genotypes were allocated to plots within watering blocks using randomised complete block designs (RCBD) with each watering block having two row-wise replicate blocks (replicates). The replicates were resolvable, with one plot of each genotype occurring in each replicate. The plots were single rows (1 m length) for which 25 seeds were sown with a custom-built stand, ensuring proper spacing between plants. Pre-seeding irrigation of 20 mm of water was initially applied across both blocks for optimal and uniform germination. Water stress treatment was applied at the stem elongation stage (BBCH scale 50) till harvest to the WD block, while stress was not applied to the WW block. Soil moisture was tracked online throughout the experiment using 1.2 m moisture probes (model EP100GL-12, EnviroPro, Rostrevor, SA, Australia) inserted in the ground and ensured that plants were subjected to water stress. Three moisture probes were installed following the manufacturer’s instructions: two in the WD block and one in the WW block; these probes were evenly spaced for reliable estimates of soil moisture across the blocks. Surface drip Irrigation was provided with a pressure-compensated line (Bunnings, Gladesville, NSW, Australia). The WD block received 6904.7 m^3^/ha of water from April to December, while the WW block was irrigated with 3333.3 m^3^/ha. At the stem elongation stage, an additional dose of urea at 75 kg ha^−1^ was applied. Post-emergence weeds were controlled manually and by applying glyphosate (Bayer Crop Science), Striker (NuFarm, Melbourne, VIC, Australia), and Revolver (Bayer Crop Science) herbicide with a shielded spray boom. Plants were protected against the blackleg fungus, *Leptosphaeria maculans* and sclerotinia stem rot fungus, *Sclerotinia sclerotiorum* by two foliar applications with Prosaro 420 SC (Bayer Crop Science, Pymble, NSW, Australia) and Aviator^®^ (Bayer Crop Science, Pymble, NSW, Australia) fungicides, as per recommendation. Broad spectrum insecticides Kensban (KENSO, Teneriffe, QLD, Australia) and Pirimidex WG (Conquest, Australia) were applied at the seed-filling stage to control aphids.

### 2.3. Trait Measurements

Four traits were measured for this experiment: days to flower (DTF), chlorophyll content with SPAD, plant height (PH) and seed yield (SY). The DTF was assessed daily for each plot and recorded when the plants had at least one open flower. Leaf chlorophyll concentration, expressed as soil-plant analytical development (SPAD) chlorophyll meter reading, was measured using a handheld SPAD (Minolta, Japan) meter. Five measurements were taken from fully expanded leaves (taken from the middle of each plant) from three randomly selected plants within each plot. The average SPAD measurements for each plot were made available for analysis. The PH was measured for five randomly selected plants in each plot. It was measured from the soil surface to the top of the inflorescence of the main stem at the physiological maturity (when pod colour changed from green to yellowish, BBCH scale (81) and was recorded in cm. The plots were hand-harvested, and the seed was cleaned with Kimseed (Australia) and then weighed in the laboratory. The SY was estimated for each plot and expressed in g. The number of plants in each plot was also recorded.

In our previous study [15], Δ^13^C measurements were available from two field experiments conducted in 2017 and 2018. Both experiments were multi-phase experiments with a field and a laboratory phase. The δ^13^C composition (^13^C/^12^C) was determined at the laboratory phase experiments conducted at the Stable Isotope Laboratory, Australian National University, Canberra, Australia, for dried leaf samples collected from field phase experiments. The δ^13^C was measured by a Micromass Isochrom mass spectrometer (Middlewich, UK). A sample of five leaves was collected when 50% of the plants in a row plot showed the first flower. Appropriate multi-phase experimental designs (see Appendix A, [14]) were used to account for the variations attributed to field and laboratory conditions [21], as described previously [22]. The experimental designs for the field phases were randomised complete block designs for both the 2017 and 2018 experiments. The laboratory phases were carried out separately for each experiment and in different ‘runs’. Each run consisted of three carousels, and each carousel contained 49 positions for which the samples were allocated, including five standards at the 1st, 2nd, 25th, 48th and 49th positions of the carousel. The Δ^13^C was determined from the leaf δ^13^C composition, measured at the laboratory phase and that of the source CO_2_ in the air (taken as −7.8‰) proposed by Farquhar and Richards [23].

### 2.4. Statistical Methods

We investigated the data for each trait separately. Based on the randomisation procedure of the factorial treatment structure (combination of the two water regimes and genotypes) described in the experimental design above, there was an aliasing of water regimes and watering blocks. Hence, there was no valid inferential framework to test the main effect of water regimes [24]. This is explained as “pseudo” or “false” replication in Bailey [25]. A valid inferential framework was used to test the genotypes by water regimes interaction (see 24). However, the aliasing of water regimes and watering blocks cautions against assigning these effects to water regimes when these effects may (at least in part) be due to watering block effects. (see Kadkol et al., [26]). Herein, we considered the two blocks as two different environments where different water regimes were applied to the two environments. Commensurate with the aims of the experiment and the structure of the datasets, single-step QTL analyses were performed on each trait using the method described in Raman et al., [15]. This method is an extension of the approach developed by Verbyla and Cullis [27] within a multi-environment trial (MET) analysis framework using factor analytic linear mixed models (FALMM) [28]. However, the QTL analyses within a MET analysis framework were only necessary for SY, PH, and SPAD traits, as DTF was measured before imposing the water regime. Hence, the QTL analysis was performed for DTF, considering that both blocks constitute a single environment. Our earlier study provides genotyping and map construction details [15]. 

All analyses were performed in *ASReml-R* [29], in which the unknown variance parameters are estimated using residual maximum likelihood (REML). These are then substituted into the mixed model equations, and the solutions are obtained for the fixed and random effects, as empirical best linear unbiased estimates (EBLUEs) and empirical best linear unbiased predictions (EBLUPs), respectively. The extent of genetic control of traits was investigated by calculating line mean heritability (*H^2^*, broad sense heritability) as the mean of the squared accuracy of the predicted DH line effects as described previously [30]. As expected, water regimes had a great influence on the environment, here the two blocks. To test the significance of GE interaction in the MET dataset for the three traits SY, PH and SPAD, we compared the models with and without the specific GE interaction using likelihood ratio tests.

We examined the genetic correlations between SY, PH, SPAD, and DTF traits within each environment (block) using multivariate analyses. For each environment, the datasets for all four traits were combined and analysed within a MET analysis framework using FALMM, where each trait was considered an ‘environment’. We also examined the genetic correlation between Δ^13^C from our previous study and SY from the current study using a multivariate analysis within the MET analysis framework using FALMM. More details on the QTL analyses and multivariate analyses are presented in Appendix A.

## 3. Results

### 3.1. Genetic Variation in DH Lines

We observed that the significant source of trait variation was genetic (marker additive genetic), which ranged from 14.58% for PH (WD block) to 75.44% for DTF (Table 1, Additive M1, %). The percentage of genetic variation accounted for by the putative QTL (Table 1, VAF_m_) ranged from 8.77 to 70% for different traits. Across watering blocks, high estimated additive and total (additive plus non-additive) genetic correlations were observed ranging from 0.70 to 1 (Table 2). In particular, the additive genetic correlations between the two blocks were very high for both PH (>0.99) and SPAD (>0.99), while a moderate correlation was observed for SY (0.7). This indicates GE interaction for SY, whereas, for PH and SPAD, there is no GE interaction. The values for the broad sense genomic heritability (*H*^2^) for each trait measured were moderate to high (Table 3) and found to be trait and or environment (block) dependent. For example, SPAD had low *H*^2^ (35%) in the WD block, but moderate values were obtained in the WW block (49%). Days to flower (DTF) had the highest *H*^2^ value (81%). Moderate to high *H*^2^ values suggest that the variation in different traits measured is heritable and, therefore, suitable for genetic analysis studies and exploiting genetic gains in the canola breeding programs. Frequency distributions of the mean values for the traits DTF, SPAD, PH and SY, assessed within the blocks among the DH lines, are presented in Figure 1. These distributions show continuous variation, suggesting that many genes control trait variation. We observed transgressive segregation beyond both parental lines for DTF, SPAD, PH, and SY among the DH lines (Figure 1). 

We further investigated the GE interaction effects across blocks using likelihood ratio tests comparing the models with and without the specific GE interaction. The *p*-values obtained from these tests revealed that there is a significant specific GE interaction present for SY (*p*-value = 0.0369) but not for PH (*p*-value = 0.8641) and SPAD (*p*-value = 0.4598). As expected, values for SY were higher under the WD block (range 63.71 to 245.77 g) compared with the WW block (range 72.80 to 415.02 g (Table 3). A significant interaction indicates a change in the scale, or ranks of genotypes, or both, between the WW and WD blocks. Among the DH population, parental lines and the check cultivars, the maximum yield reduction occurred for BC1329 (40.8%), the maternal parent, and 06.5101.411 (40.3%). In contrast, the minimum yield reduction was for the 06.5101.059 DH line (12.5%) (Figure 2). These results also showed that the maternal parent, BC1329, is more sensitive to drought stress than the check cultivars, Ag-Spectrum, Skipton, Charlton, Tarcoola, and Monty (33.8 to 39.5% yield reduction) (Figure 2). 

### 3.2. Genetic Correlations between Traits for the Two Blocks

Genetic correlations between traits were investigated within WW and WD blocks among the DH lines and presented in Figure 3. There was a strong negative correlation between DTF and SY at the WW block (*ρ* = −0.51). Irrespective of water regimes, SPAD was negatively correlated with SY, while a positive correlation was observed between PH and SY (Figure 3). The DTF and SPAD values at the WW block were positively correlated (*ρ* = 0.41). We also investigated the genetic correlation between SY from the WW block of the current study and leaf Δ^13^C from two field trials of our previous study [15]. Aligning with the previous study [15], a positive genetic correlation was observed between these two traits (*ρ* = 0.34).

### 3.3. QTL Analysis Identifies QTL for Main Effects and QE Interaction Effects for Productivity Traits in Contrasting Water Regimes

Our analyses identified genomic regions having QTL main effects and QE interaction effects for a set of four traits associated with SY. We identified 23 QTL for the main effects on chromosomes A01, A02, A03, A08, A09, A10, C02, C03, C05, C06, and C09, accounting for 3.36 to 48.55% of the genetic variance (Table 4). The QTL for QE interactions were only identified for SY (A02 and C09) as there was no specific GE interaction for PH and SPAD—the only traits measured after imposing water regimes.

Eight significant QTL for main effects were identified for DTF on chromosomes A01, A08, A09, A10, C02, C06, and C09 (Table 4, Figure 4a). Each QTL explained an average of 23.27% (ranging from 9.11 to 48.55%) of the genotypic variance. Both parental lines contributed allelic effects for variation in DTF. The BC1329 alleles delayed flowering by 5.06 days, while BC9102 alleles promoted flowering time up to 2.24 to 4.13 days (Table 4). 

Seven QTL for main effects were identified for SPAD on chromosomes A01, A02, A03, A09, A10, CO5, and C09 (Table 4, Figure 4b). The QTL on A01 and A03 had a LOD score of less than 3 and therefore were designated as minor QTL. Each QTL explained 9.44% to 11.05% of the genetic variance. Both parental lines contributed allelic effects for variation in SPAD, consistent with the phenotypic frequency distribution (Figure 1c). The BC9102 alleles reduced chlorophyll content by 0.78 to 1.3 units, while the BC1329 alleles increased chlorophyll content up to 1.01 units (Table 4).

For PH, three QTL for main effects were identified on chromosomes C03 and C09 (Table 4, Figure 4b). These QTL accounted for 3.36% to 6% of the genetic variance. Only the maternal parent, BC1329, contributed alleles for increased PH. 

Five QTL (three for main effects and two for QE interactions) were detected for SY (Table 4, Figure 5). The QTL for main effects were identified on A08, A09, and C03, while the QTL for QE interactions were identified on chromosomes A02 and C09. These QTL had lower LOD values (2 to 2.76) but were repeatedly detected in two contrasting water regimes. Interestingly, both parental lines (BC1329 and BC9102) contributed alleles for SY variation at these genomic regions in opposite directions (Table 4). For example, in a QTL on A02 (peak marker 3173313 mapped to 19.73 Mb), the BC9102 allele decreased SY under WW conditions, while the BC1329 allele increased SY under WD conditions (Table 4). Opposite allelic effects were also observed at the QTL on C09 (peak marker 5121657 mapped to 41.79 Mb), where the BC1329 allele increased the SY while the BC9102 allele decreased the SY. The SY QTL accounted for 0.74 to 4.61% of genetic variation (Table 4). Cumulatively, the SY QTL accounted for 8.77% of the genetic variance (Table 1, VAF_m_). The rest of the genetic variation was not accounted for by markers, suggesting that SY is a complex trait with a moderate *H^2^* (58 and 59% for WD and WW blocks, respectively, Table 1). Two QTL were associated with multiple traits (Table 4, Figure 5); the first QTL for DTF and SY was on chromosome A09 and the second for PH and SPAD on C09 (Figure 5). These QTL were mapped to the same genomic region, i.e., the marker 3153720 localised on A09 to 29.3 Mb sequence of Darmor-*bzh* and DArTseq-SNP marker 3101645|F|0–42:G>A-42:G>A/27247510 on chromosome C09 (~48.14 to 48.49 Mb). 

### 3.4. Comparison of QTL across Environments

To determine the stability of the QTL for SY and related traits across phenotypic environments, we compared the physical positions of significantly associated markers identified in this study (Appendix A) and compared with those which were identified earlier in the same DH population in our previous study [15]. One QTL on chromosome A09 was co-located and repeatedly detected across environments: the main effect QTL for field experiments conducted in 2017 and 2018 and pot experiment conducted under a rainout shelter in 2017 (previous study); the main effect QTL for WW and WD blocks in a rainout shelter in 2018 (current study, Figure 5). This QTL delimited with the DArTseq 3153720 marker on chromosome A09 showed a significant association with Δ^13^C, DTF, PH, and SY (14, Appendix A). 

For DTF, five QTL on A01, A08, A09, A10, and C09, identified in the current study were localised within 863 kb from genomic regions identified previously for the QTL main effects for DTF, PH, Δ^13^C and QE interaction effects for DTF (Appendix A). For PH, only one QTL on C09 identified in the current study was mapped 677 kb apart from that was detected for the PH QTL for QE interaction in our previous study (Appendix A). For SY, besides the A09 QTL delimited with the 3153720 marker, two other QTL on A08 (with peak marker 3140140) and C03 (associated with marker 3101614|F|0–46:C>T-46:C>T) identified in the current study were localised approximately 720 kb and 107 kb, respectively, apart from the SY QTL for QE interaction identified in our previous study (Appendix A). The QTL detection for QE interaction suggested that these loci related to plasticity in different water regimes. 

## 4. Discussion 

Breeding for drought tolerance is one of the major objectives of global canola breeding programs. Several yield-related traits, such as plant vigour, biomass, flowering time, transpiration efficiency and root architecture, play an important role in drought resistance mechanisms and influence seed yield [11,13,14,15,22]. Therefore, the present research was carried out to understand the genetic basis of seed yield and its related traits in contrasting water regimes. We focussed on a DH population derived from an F_1_ cross between BC1329 (maternal parent) and BC9102 (paternal parent). This population has revealed genetic variation for a range of traits related to plant development (shoot/root biomass and NDVI), phenology (flowering time and plant height), effective water use (Δ^13^C, *A*, g_sw_, *i*WUE, TE), harvest index (HI), and seed yield [15,20]. In addition, genomic regions contributing to the QTL main effects and QE interaction effects for various effective water use traits such as Δ^13^C, NDVI, days to flower, plant height and seed yield were also identified in this DH population grown across three environments [15]. Leveraging the available phenotypic and trait-marker association data, we set to determine the response of water-deficit stress on seed yield and related traits under semi-controlled rainout shelter conditions. Under these conditions, the application of water stress can be assured, unlike field conditions, where there is no control for natural rainfall. The application of water stress treatment to accessions at the same phenological stage under rainout shelter and field conditions is not practical. Herein, we applied water-deficit treatment at the stem elongation/pre-flowering stage to identify the QTL main effects and QE interaction effects.

### 4.1. Phenotypic Plasticity to Water-Deficit Conditions

Our analysis suggests that WD conditions significantly impact the seed yield, possibly affecting source-sink relations. Water is essential for various plant development, physiological, metabolic, and reproduction-related activities. However, under WD conditions, plants evolved by deploying various drought avoidance and tolerance strategies for their survival and reproductive success. This was evident from trait plasticity; different genotypes responded variably, especially under WD conditions, suggesting that genetically controlled variation for drought adaptation strategies exists among DH lines, including parental lines (Table 4). Two QTL on chromosomes A09 and C09 showed significant QE interaction effects, indicating that there are loci that contribute to physiological plasticity between WW and WD environments. Favourable alleles of BC9102 detected under both WW (SY on C09), and WD (SY on A02) conditions could be combined to improve canola yield. Furthermore, early flowering lines yield higher than late flowering genotypes, even in WW conditions, suggesting drought escape trait; early flowering does not pose any yield penalty to late ones. 

### 4.2. Potential Proxy Traits for Selection of Genotypes with Improved Water Productivity

The selection of canola varieties for high seed yield in optimal environments has been very successful and has increased production worldwide. However, only a few traits, such as early flowering and vigour, have been intentionally selected in the breeding programs to target canola cultivars suitable for cultivation in terminal WD conditions. We attempted to identify and validate proxy traits for TE (such as chlorophyll content with SPAD) that can be used to select improved TE and high SY. We observed negative correlations between SY and SPAD; and a positive correlation with DTF (r = 0.41, Figure 3a). These results suggest that genotypes with low SPAD, early flowering, and higher PH likely improve SY, irrespective of water regimes. Recent studies showed that low chlorophyll lines exhibit similar or high gross carbon uptake and biomass accumulation [31,32,33,34]. Previous studies have also shown a negative correlation between variation in flowering time and related seed yield [10,13,15,22,35,36], as observed herein (Figure 3). Our results supported the previous findings and inference that days to flower is an upstream determinant of seed yield [13]. Moderate to high correlations between DTF, SPAD, PH, and SY, suggesting that correlated traits can be used for indirect selection to make a genetic gain in seed yield. The correlation may exist because of a similar physiological mechanism or pleiotropy i.e., due to the genetic linkage of loci affecting different traits.

### 4.3. Identification of Stable QTL for Seed Yield and Related Traits

Earlier studies have identified several seed yield-related QTL on different chromosomal positions in *B. napus* mapping and diversity panels [37]. However, the QTL associated with seed yield under contrasting water regimes were not described. In the present study, three genomic regions for QTL main effects were identified for seed yield on chromosomes A09, C03 and C09, while two were related to QE interactions on A02 and C09. Opposite allelic effects on plasticity in seed yield were identified due to QE interactions in response to water stress on chromosomes A02 and C09. At least one QTL on chromosome A09 was also localised within 0.5 Mb (*BnaA0950250D*) in other *B. napus* populations [38]. Our results showed that the QTL allelic effects for DTF, PH, SY and their correlation with Δ^13^C are stable across environments (rainout shelter and field conditions [15]) and contrasting water regimes (this study). 

In the present study, we identified six QTL for variation in SPAD (chlorophyll content) on A01, A02, A03, A09, A10, and C09 chromosomes (Table 4). These results suggested that multiple genes determined chlorophyll content. Quantitative inheritance was also evident from the continuous distribution of SPAD values (Figure 1). At least one of the QTL for SPAD (Table 4) was also mapped on A02 to an interval of 21.87–22.91 Mb (designated as *cqSPDA2)* in the backcross population of *B. napus* derived from genotypes: QU (deep green leaves, high-chlorophyll content, SPAD value = 50.4) and ZS11 (light green leaves, low-chlorophyll content, SPAD value = 40.6) [39]. The authors further narrowed the mapping interval to a 152 kb region and prioritised three annotated *B. napus* genes, BnaA02g30260D, BnaA02g30290D, and BnaA02g30310D, which may be responsible for chlorophyll synthesis [39]. Further research is required to determine whether the same set of genes controls variation in chlorophyll content in the DH population from BC1329/BC9102. Our QTL mapping results did not conform with an earlier study that detected a genomic region on chromosome C08 using chlorophyll-deficient *B. napus* mutants [40]. 

Up to 70 QTL for plant height on 15 chromosomes have been reported in *B. napus* [41,42,43]. The QTL on C09 that we identified (Table 4) was also detected in a population derived from a cross between Y689 (*Capsella bursa-pastoris* derived *Brassica napus* intertribal introgression line) and *B. napus* cultivar, Westar [44]. Our results were inconsistent with finding made in a GWAS panel of 472 rapeseed accessions that identified eight QTL on chromosomes A03, A05, A07, and C07 [43]. Given the quantitative nature of drought tolerance, correlated traits could be useful for selection to make continued genetic gains in breeding programs. Two QTL regions, one for plant height and chlorophyll content on chromosome C09 and the second for flowering time and seed yield on chromosome A09, were co-localised to similar genomic regions. This A09 QTL was also detected in a previous study for Δ^13^C that showed a positive relationship with seed yield in the same population [15]. The SPAD QTL was also mapped in the vicinity of marker 26680018, localised on 26.4 Mb (Table 4). One of the QTL detected for drought avoidance trait (root pulling force) on A10 that appeared in both wet and dry treatments was mapped to the flowering time gene (FLOWERING LOCUS C, BnaA10g22080D) in the DH population derived from IMC106RR and Wichita [13,16]. As identified in this study, the co-localisation of QTL for multiple traits could be considered circumstantial evidence for pleiotropy [45,46]. In addition, several QTL detected on C02 (DTF), C09 (PH), A03, and A10 (SPAD) were mapped near the FLOWERING LOCUS C orthologues in our low-density mapping. Our results on QE interaction for plasticity to terminal WD conditions are based on one year and need validation. However, single-season phenotyping for identifying drought tolerance genotypes has also been reported in soybean [47]. Rainout shelters provide a similar environment to the glasshouse and more control over water regimes under field conditions. Further research is required to determine the precise location of flowering time genes and their relationship with drought tolerance, plant growth, and yield-related traits. Nevertheless, earlier studies have also detected GWAS and QTL signals for pleiotropy effects for days to flower, plant growth and yield-related traits [48,49]. 

In summary, under contrasting water regimes, we identified 23 QTL associated with main effects and QE interaction effects for flowering time, SPAD, plant height, and seed yield. Seven QTL had LOD scores less than 3 and were categorised as minor. At least one stable QTL was identified on chromosome A09 for flowering time and seed yield via multi-environment QTL analyses, suggesting that this QTL represents a locus having pleiotropic effects on multiple traits. This is supported by the high correlation of SY with DTF and PH. In a recent study, Raman et al. [15] also identified the same QTL associated with multiple traits across the field and rainout shelter environments. Detecting QTL on chromosome A09 across the field and rainout shelter environments with contrasting water regimes suggests that it can improve water productivity in canola breeding programs. Favourable alleles for QTL main effects and QE interaction effects can be enriched to breed new canola cultivars for target environments. Associated SNP markers may provide tools to incorporate favourable alleles to improve seed yield. 

## 5. Conclusions

This study identified 23 QTL (16 with significant QTL main effects and QE interaction effects and 7 with minor effects) distributed on A01, A02, A03, A08, A09, A10, C02, C03, C05, C06, and C09 chromosomes for chlorophyll content, flowering time, plant height, and seed yield. Proxy traits and linked markers associated with drought tolerance could enable the indirect selection of improved canola yield. One major QTL on chromosome A09 that accounted for 4.23% to 17.81% of the genotypic variance for multiple traits (leaf Δ^13^C, DTF, PH, SY) was detected in four environments (rainout shelter, and field) and contrasting water regimes. The SNP markers associated with QTL main effects and QE interaction for phenotypic plasticity will enable canola breeding programs to make a genetic gain for seed yield across well-watered and water-deficit environments.

## Figures and Tables

**Figure 1 plants-12-00720-f001:**
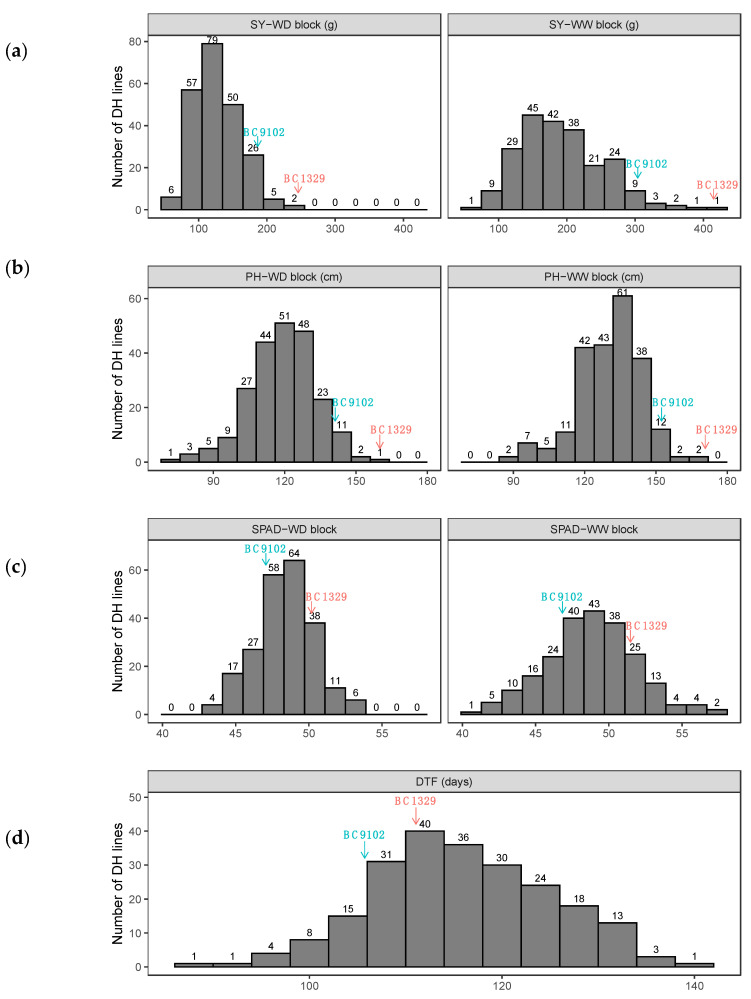
Frequency distributions of the traits for 223 DH lines from the BC1329/BC9102 breeding population grown in two contrasting water regimes: well-watered (WW) and water-deficit (WD) conditions under a rainout shelter at Wagga Wagga Agricultural Institute. Estimates for the parental lines are shown with arrows. (**a**) SY: Seed yield; (**b**) PH: Plant height; (**c**) SPAD: Chlorophyll content; (**d**) DTF: Days to flower. Total (additive plus non-additive) common genotype by environment empirical best linear unbiased predictions are used for the traits SY, PH, and SPAD, and total genotype empirical best linear unbiased predictions are used for DTF for the frequency distributions.

**Figure 2 plants-12-00720-f002:**
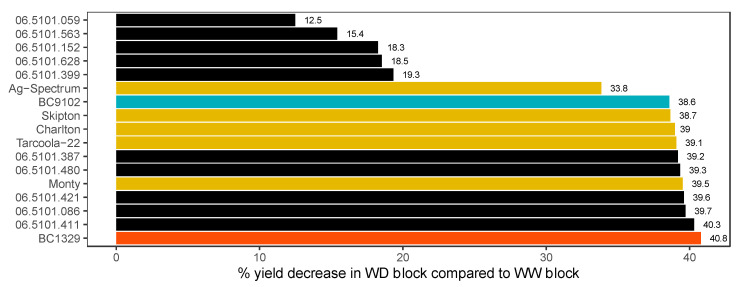
Per cent seed yield decrease of the water-deficit (WD) block compared to the well-watered (WW) block for check cultivars, parental lines and some high and low per cent seed yield decreased DH lines of interest.

**Figure 3 plants-12-00720-f003:**
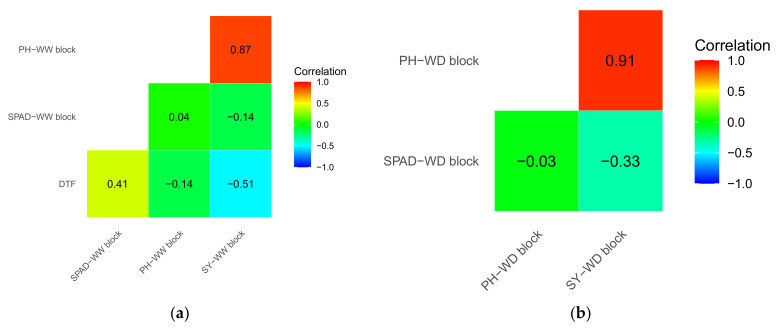
Genetic (total, additive plus non-additive) correlations between traits (**a**) within well-watered (WW) and (**b**) within water-deficit (WD) blocks from the multivariate analyses. SY: Seed yield; PH: Plant height; SPAD: Chlorophyll content; DTF: Days to flower.

**Figure 4 plants-12-00720-f004:**
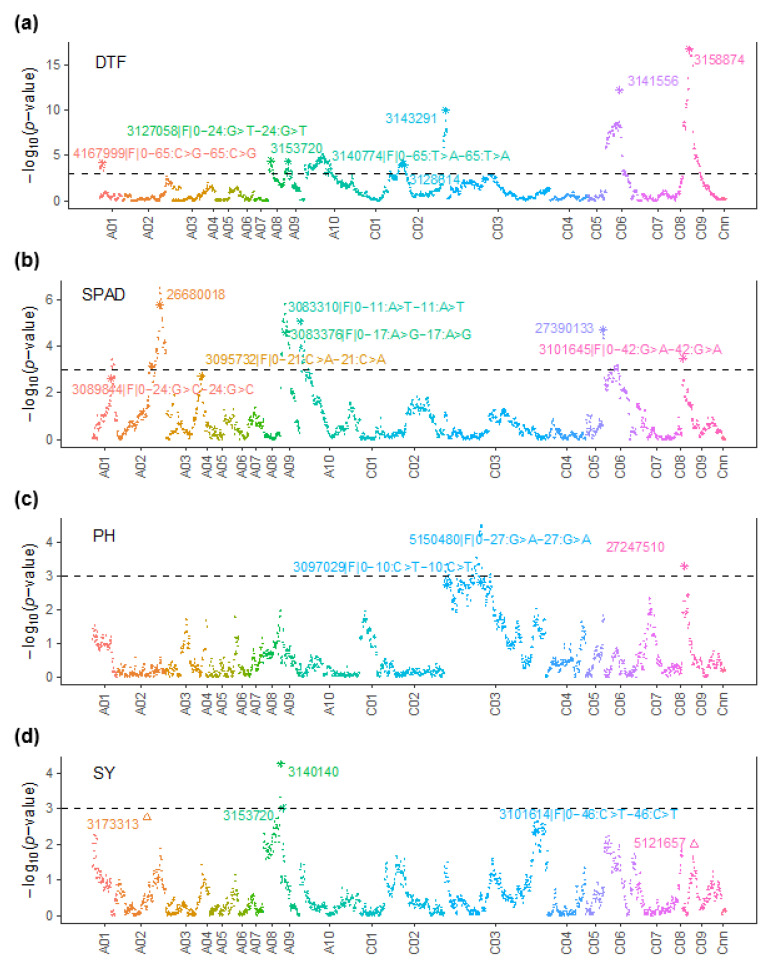
Manhattan plots showing genomic regions associated with (**a**) DTF: Days to flower; (**b**) SPAD: Chlorophyll content; (**c**) PH: Plant height; and (**d**) SY: Seed yield, in the DH population. The QTL for main effects are depicted by ‘*’, and the QTL by Environment interactions are depicted by ‘Δ’. The LOD (-log_10_*p*-value) scores presented in the Manhattan plot are from the genome scan for the QTL main effects where the LOD scores of the significant QTL are replaced with the values from the final model. The black dashed line indicates the threshold value for significant SNPs at LOD ≥ 3. The physical positions of DArTseq markers (*x*-axis) are based on the map position on the Darmor-*bzh* genome assembly (for detail, see Appendix A).

**Figure 5 plants-12-00720-f005:**
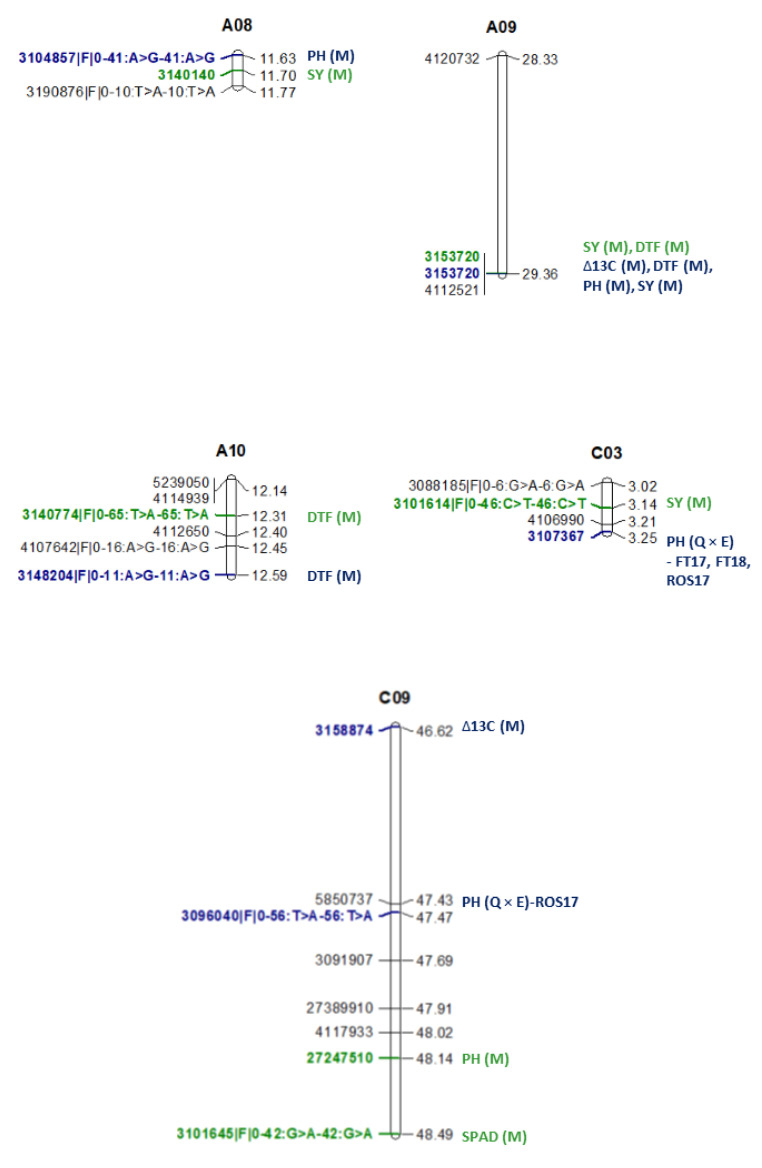
Graphical representations of QTL associated with multiple traits in a doubled haploid population of *B. napus*. Map distances (Kosambi) are shown in CentiMorgans (on right). Marker names are shown on the left-hand side. SY: Seed yield, PH: Plant height; SPAD: Chlorophyll content; DTF: Days to flower; Δ^13^C: carbon isotope discrimination. QTL for main effects are depicted by M, and QTL by Environment interactions are depicted by ‘Q × E’. Field trial 2018: FT 18, Field trial 2019: FT19; pot trial in a rainout shelter 2017: ROS17.

**Table 1 plants-12-00720-t001:** Summary of the partitioning of the genetic variance into Additive and Non-additive and the residual maximum likelihood estimates of the Total (additive plus non-additive) genetic variance before (Baseline factor analytic linear mixed model with markers: M1) and after identifying putative QTL (Final multi-QTL model: M2) for each of the traits. VAF_m_ shows the percentage of genetic variance accounted for by the identified putative QTL. Trait variations for SY, PH, and SPAD were assessed considering the two watering blocks as two environments, whereas for DTF, it was evaluated considering them as a single environment; SY: Seed yield, PH: Plant height; SPAD: Chlorophyll content; DTF: Days to flower; WD: Water-deficit block; WW: Well-watered block. Seed yield is predicted at the average value of 9.6 plants per plot for WW block and 6.3 plants per plot for WD block.

Trait	Environment	Additive		Non-Additive		Total		VAF_m_ (%)
		(M1, %)	(M2, %)	(M1, %)	(M2, %)	(M1)	(M2)	
SY	WD	19.70	6.99	80.30	93.45	1558.57	1242.56	8.77
	WW	18.33	14.55	81.67	86.35	4304.96	4106.61	
PH	WD	14.58	1.10	85.42	98.94	233.00	208.55	15.83
	WW	21.88	0.05	78.12	99.95	214.68	168.28	
SPAD	WD	68.61	75.04	31.39	29.02	9.40	3.05	60.27
	WW	57.16	30.72	42.84	70.94	15.69	6.92	
DTF	-	75.44	35.61	24.56	64.39	118.77	35.63	70.00

**Table 2 plants-12-00720-t002:** Residual maximum likelihood estimates of the between environment (watering block) genetic correlations for Additive and Total (additive plus non-additive) effects of each trait from the baseline factor analytic linear mixed model with markers. Correlations are presented only for SY, PH, and SPAD, for which the two blocks were considered as two environments in the analysis; SY: Seed yield, PH: Plant height; SPAD: Chlorophyll content; WD: Water-deficit block; WW: Well-watered block.

Trait	Effects	Correlation
SY	Additive	0.70
	Total	0.94
PH	Additive	>0.99
	Total	>0.99
SPAD	Additive	>0.99
	Total	0.88

**Table 3 plants-12-00720-t003:** Summary of heritability, mean, minimum and maximum values for each trait in the doubled haploid population from BC1329/BC9102 together with their parental lines. SY: Seed yield; PH: Plant height; SPAD: Chlorophyll content; DTF: Days to flower; WD: Water-deficit block; WW: Well-watered block. Total (additive plus non-additive) common genotype by environment empirical best linear unbiased predictions are summarised for the traits SY, PH, and SPAD, and total genotype empirical best linear unbiased predictions are summarised for DTF.

Trait	Environment	Heritability	Minimum	Mean	Maximum	BC1329	BC9102
SY (g/row plot)	WD	0.58	63.71	127.21	245.77	245.77	186.59
	WW	0.59	72.80	192.20	415.02	415.02	303.78
PH (cm)	WD	0.66	75.34	119.08	159.98	159.98	141.21
	WW	0.69	87.80	130.60	170.70	170.70	152.32
SPAD (unit)	WD	0.35	43.16	48.40	53.73	50.19	47.06
	WW	0.49	41.04	48.83	56.74	51.48	46.83
DTF (days)	-	0.81	89.78	115.99	138.16	111.06	105.74

**Table 4 plants-12-00720-t004:** Markers associated with the QTL main effects and QTL by Environment (watering block) interaction for seed yield (SY) and other agronomic traits (DTF: Days to flower; PH: Plant height; SPAD: Chlorophyll content) using a doubled haploid population derived from BC1329/BC9102, in well-watered and water-deficit conditions under a rainout shelter (M: main effect; WW: well-watered block; WD: water-deficit block). The LOD scores (−log_10_
*p*-value), allelic effect, parental allele and percentage of genetic variance explained (*R*^2^) are also provided. The QTL by Environment interactions for each environment are in italics. Consistent markers that were associated with multiple traits are in bold font. The * QTL for corresponding traits were detected under field trials and a pot trial in rainout shelter conditions [15]; ≠Marker positions are approximate, as the DArTseq did not return significant hits for desired linkage group. NA: Unknown (marker sequence could not be mapped onto the reference sequence).

Trait	Environment	Marker	Chromosome	Physical Map Position of ‘Top’ Marker on Darmor-*bzh* Genome Version 4.1	LOD	*R* ^2^	Allelic Effect	Parental Allele
DTF	M	4167999|F|0–65:C>G-65:C>G	A01	4,038,480	4.15	9.11	2.41	BC1329
DTF	M	3127058|F|0–24:G>T-24:G>T	A08	NA	4.41	14.26	−2.24	BC9102
***DTF**	**M**	**3153720**	**A09**	**29,356,333**	**4.31**	**17.81**	**2.19**	**BC1329**
DTF	M	3140774|F|0–65:T>A-65:T>A	A10	NA	3.15	17.58	−2.44	BC9102
DTF	M	3128614	C02	9,287,096	3.88	13.04	−2.25	BC9102
DTF	M	3143291	C02	45,636,489	9.95	26.7	−3.31	BC9102
DTF	M	3141556	C06	27,740,738	12.17	39.12	−4.13	BC9102
DTF	M	3158874	C09	46,623,311	16.66	48.55	5.06	BC1329
PH	M	5150480|F|0–27:G>A-27:G>A	C03	23,396,698	2.82	3.36	3.46	BC1329
PH	M	≠3097029|F|0–10:C>T-10:C>T/ 5034370|F|0–47:G>C-47:G>C	C03	NA 57,776,378	2.75	4.68	3.34	BC1329
**PH**	**M**	**27247510**/**≠3088657**	**C09**	**NA**/**48,143,335**	**3.3**	**6**	**3.76**	**BC1329**
SPAD	M	3089844|F|0–24:G>C-24:G>C	A01	7,826,453	2.61	11.05	0.77	BC1329
SPAD	M	26680018	A02	23,433,061	5.79	27.15	−1.3	BC9102
SPAD	M	3095732|F|0–21:C>A-21:C>A	A03	1,882,135	2.72	9.44	−0.78	BC9102
SPAD	M	3083376|F|0–17:A>G-17:A>G	A09	26,477,098	4.58	16.46	1.01	BC1329
SPAD	M	3083310|F|0–11:A>T-11:A>T	A10	16,253,199	5.06	24.39	−1.14	BC9102
SPAD	M	27390133	C05	539,869	4.72	19.51	−0.98	BC9102
**SPAD**	**M**	**3101645|F|0–42:G>A-42:G>A/** **≠3156841**	**C09**	**NA** **48,490,657**	**3.48**	**14.94**	**0.87**	**BC1329**
*SY*	*WD*	*3173313*	*A02*	*19,738,340*	*2.76*	*1.13*	*3.91*	*BC1329*
*SY*	*WW*	*3173313*	*A02*	*19,738,340*	*2.76*	*1.13*	*−12.86*	*BC9102*
SY	M	3140140	A08	11,695,725	4.26	1.87	14.38	BC1329
***SY**	**M**	**3153720**	**A09**	**29,356,333**	**3.02**	**4.23**	**−11.92**	**BC9102**
SY	M	3101614|F|0–46:C>T-46:C>T	C03	3,138,929	2.35	4.61	9.94	BC1329
*SY*	*WD*	*5121657*/≠3079649	*C09*	*NA*/41,790,279	*2*	*0.74*	*−3.71*	*BC9102*
*SY*	*WW*	*5121657/≠3079649*	*C09*	*NA/41,790,279*	*2*	*0.74*	*9.99*	*BC1329*

## Data Availability

All data generated or analysed in this study are available in the manuscript and the supplementary materials. Seeds of the populations are available for research and commercial applications via a material transfer agreement/licensing agreement from the corresponding author (H.R.).

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
