# Peer review of "Quantitative Trait Loci for Genotype and Genotype by Environment Interaction Effects for Seed Yield Plasticity to Terminal Water-Deficit Conditions in Canola (Brassica napus L.)"

_plants, 2023, doi:10.3390/plants12040720_

Round 1

Reviewer 1 Report

The manuscript describes quantitative trait locus (QTL) mapping analysis of genetic regions that influence days to flower, plant height, chlorophyll content and seed yield in canola plants, when exposed to water deficit conditions. Canola accessions analysed exhibited genetic variation in traits involved in drought escape and drought avoidance strategies.

The study is interesting and is very well written. However, several suggestions are made to improve the manuscript.

Line 39 – Abbreviation DH was not explained.

Figure 1  - Please insert coordinates of y axis

111 – Is not clear what are the characteristics of Ag-Spectrum, Charlton and the other cultivars that are interesting to the generation of the 223 lines under study.

Line 464 – Figure 5a was not shown.

Figure 5 – Figure 5 lacks quality and the information is difficult to read.

Line 465 – Chromosome A02 was not shown.

Line 493 – Supplementary Table 2 was not submitted to Plants Journal.

Line 551 - cultivars (change misspelled word)

Author Response

Thank you very much for your comments. We have addressed your comments (please see edits in TRACK CHANGES).

Line 39 – Abbreviation DH was not explained.

Line 39: We have inserted DH in brackets.

Figure 1  - Please insert coordinates of y axis

We have already provided this (DH lines, represents to 'number')

111 – Is not clear what are the characteristics of Ag-Spectrum, Charlton and the other cultivars that are interesting to the generation of the 223 lines under study.

These varieties were used as CHECKS, in addition to 223 DH lines (please line 142).

Line 464 – Figure 5a was not shown.

Sorry this was a mistake. We have deleted this (please see line 473).

Figure 5 – Figure 5 lacks quality and the information is difficult to read.

We have revised this Figure.

Line 465 – Chromosome A02 was not shown.

Line 493 – Supplementary Table 2 was not submitted to Plants Journal.

Line 551 - cultivars (change misspelled word)

Thank you for correcting this. We have gone through the manuscript and made corrections (please see track changes)

Author Response

Letter addressing comments-Reviewer 2

Thank you very much for your insights in statistical genetics and positive feedback. We have addressed your comments (written in bold better after your comments).

Regards, Harsh

 The authors report on an experiment conducted on a Doubled Haploid population of canola, and present a sophisticated statistical analysis of a multi-environment QTL model for seed yield, DTF, SPAD and PH. While the experiment used in this study has been reported before, together with additional traits and additional experiments/environments (Raman et al, 2022), the authors clearly state in the introduction that the aim of this paper was to investigate the “plasticity of seed yield and its related traits in response to water stress at flowering….” and to compare the detected QTL for main effects and interaction with those identified in earlier studies.

The authorship team clearly show advanced skills and knowledge across multiple disciplines including physiology, plant breeding, genomics and statistics, and it is refreshing to review a study from a multi-disciplinary approach incorporating a powerful statistical analysis and elegant presentation and interpretation of results.

However, I do have a major reservations about the overall research design as being fit-for-purpose as a GxE or, more specifically, a QxE study. Only one experiment in one year is used for this QxE study, where environment is defined by water and water-deficit treatments that are formed through pseudo-replication. While controlled conditions define these environments, the experiments are grown in the field (under shelters) and there is no repeat of the GxE study across multiple environmental conditions induced by differing temperature, humidity, daylength, etc. and other factors that are represented by seasonal and geographic variation that normally occur in a multi-environment trial setting. For this reason, the QxE study is limited to a single contrast between water regimes in one experiment, and this single scenario may not be representative of all environmental conditions under which this contrast should be assessed. For this reason, I believe there is a severe limitation with the detection and interpretation around both the main effect and stability of QTL across environments. Consequently, the interpretation of results should be modified to highlight the limitations of these findings, given the research design.

We agree with the reviewer’s comment. We have addressed this comment raised by reviewer 1.  We have highlighted this limitation in the ms. Single environment-based selection for drought tolerance has been reported in several papers. For example, please see 662-665

The second major reservation is that seed yield is measured in a single row plot of 1m in length. Much larger plot sizes are required for reliable yield estimates in the field, and the detection of QTL from such small plots may be misleading, regardless of statistical significance. In addition, it is well-known that inter-plot interference occurs between single-row plots and this may also be impacting the genetic estimates for seed yield. If the inference from this study relies on these seed yield values being representative of genetic performance for grain yield (t/ha) in larger plots or in a pure stand in farmers’ fields, then the study is compromised.

We agree with this comment fully. Rainout shelters are a valuable resource and can be used for screening a limited set of germplasm. There is no such rainout facility worldwide that can accommodate a mapping population of 225 lines x 2 replicates x 2 treatments. In Australia, we sow large plots (10 m long, later on, cut back to 8 m for agronomic management x 2 m wide) with our farm machinery.

 In our previous study, we phenotyped this population under field plots and in pots. There were high trait correlations. Please see our previous paper (Raman H , Rosy RamanRamethaa PirathibanBrett McVittieNiharika SharmaShengyi LiuYu QiuAnyu ZhuAndrzej KilianBrian CullisGraham D FarquharHilary Stuart-Williams , Rosemary White , David Tabah , Andrew EastonYuanyuan Zhang  (2022) Multienvironment QTL analysis delineates a major locus associated with homoeologous exchanges for water-use efficiency and seed yield in canola. Plant Cell and Environment 45(7):2019-2036.  doi: 10.1111/pce.14337).

Secondly, several QTL detected in study for DTF (chromosomes A01, A09, C06 and C09), PH (on C09) and SY (on A08, A09 and C09) were mapped to the same genomic regions on the Brassica napus genome assembly which were detected in field plots (Plant Cell and Environment 45(7):2019-2036.  doi: 10.1111/pce.14337). This suggests that QTL mapped based on single-row plots is not unreliable.

There are several studies published in high-quality journals like the Journal of Experimental Botany (Mohamed El-SodaMartin P. BoerHedayat BagheriCorrie J. HanhartMaarten KoornneefMark G. M. Aarts  (2014) Genotype–environment interactions affecting pre-flowering physiological and morphological traits of Brassica rapa grown in two watering regimes. Journal of Experimental Botany, Volume 65, Issue 2, February 2014, Pages 697–708, https://doi.org/10.1093/jxb/ert434)) which used even a single plant/pot in three replications and identified QTL for genotype x environment interaction.

General comments

The manuscript in its current form is quite verbose and requires a major rewrite of some sections. The first paragraph of the Introduction requires polishing as in does not provide a smooth flow to introduce the background to this study. The statistical analysis section reads somewhat like a text book, and for journal articles this should be substantially reduced to contain only a description of the methodology used in the analysis.

We have rewritten this section (please in track changes). Thank you for your comments

There appears to be only one file in the Supplementary material containing a photograph of the rainout shelters. Unfortunately, the further details on the analysis and results was not present in the Supplementary material.

Sorry for this; some of the files were not loaded on the website. We have uploaded them (please supplementary files.

Specific edits required are:

Title: edit Condi-Tions

We have corrected this

Line 129: Do you mean water-watered conditions?

We have fixed this typo (please see line 178)

Line 141: No Supplementary Figure 2 supplied.

We have supplied this

Line 200-201: spelling of carousels/carousals.

We have fixed this tyo (please see line 245 )

Line 243: No Supplementary method S1 supplied.

We have supplied this S1.

Lines 234 and 248: suggest the authors use either GxE or GE for genotype by environment, but not both. This occurs throughout the text.

We made changed to GE on sveeral places. Please see track changes.

Lines 310-313: Results, not Methods.

We have corrected this

Lines 324-326: How was the K matrix calculated in the pedicure package, as it is not simply MM’. It is important to specify the centering and scaling options if you are claiming to use the vanRaden (2008) form of K. Also, the matrices should appear in bold form if you are following standard statistical notation.

We have made suggested changes. Please supplementary methods.

Table 2: There are only six correlation values in this Table (that is, one number from the 2x2 table of 1’s and correlations) and I suggest these could be presented in a much simpler format.

We have revised Table 2, as suggested.

Line 493: Supplementary Table 2 is not supplied.

We have supplied Table S2.

Reviewer 3 Report

In the presented manuscript, the authors identified 23 QTLs with effects in terms of chlorophyll content, flowering time, plant height and seed yield. Surrogate traits and associated markers related to drought tolerance may enable improved breeding for improved oilseed rape yield. 

One major QTL counted for 4.23% to 17.81% of the genotypic variance for multiple traits was detected in four environments and contrasting 

water regimes. SNP markers associated with interaction for phenotypic plasticity will enable canola breeding programs to make a genetic gain for seed yield across well-watered and water-deficit environments.

The paper is written correctly, the statistical methods and controls selected are appropriate. It is noteworthy that the paper describes very interesting plant material.  The only comment is on editing errors.

Author Response

We have corrected editing errors. Please see the revised MS. Regards

Reviewer 4 Report

The authors have made great efforts to evaluate the plasticity of seed yield and its related traits in response to water stress imposed at the flowing time in a DH population. The attained data are of interest to the readers and agronomists to further improve canola breeding programs under water-deficit environments. The comments and suggestions may improve the quality of the current form of the manuscript

1. The title, abstract should add the scientific name of canola

2. Many abbreviated words should be fully written in advance, then made abbreviation, check whole manuscript

3. Some words should be reworded in uniformity, for instance "rainout" or "rain-out" etc.,

4. The goals of this study should be rewritten for clearer based on the research contents of this study.

5. In Materials and Methods part, the five check cultivars must be mentioned why they were used.

6. The experimental design is the most important section, hence, well-water and water-deficit must be narrated in details.

7. Subsection 2.4.2 " Investigation of G x E interaction in the MET dataset", no cited reference was used, therefore, it should mention as the methodology developed by the authors.

8. The result sections were well written, however, Fig 5 should be replaced by another one with high quality and solution.

9. One of the weakest point of this study was to the experiments were done one only under rainout shelters, hence the attained results may not be reliable. Therefore, the discussion must be mentioned this points in detail.

10. All cited references must be listed in number following the style of Plants journal

Author Response

Thank you very much for the comments. Here is our response 

  1. The title, abstract should add the scientific name of the canola 

We have included the scientific name of canola (Brassica napus L.) in the title. Please see the revised manuscript in track changes (line 4).

2. Many abbreviated words should be fully written in advance, then made abbreviation, check whole manuscript

We have checked the manuscript and made suggested changes.

3. Some words should be reworded in uniformity, for instance "rainout" or "rain-out" etc.,

We have corrected the uniformity now.

4. The goals of this study should be rewritten for clearer based on the research contents of this study.

we have reworded the goal of this study (lines 105-106)

5. In Materials and Methods part, the five check cultivars must be mentioned why they were used.

We have described the check cultivars (lines 119-120)

6. The experimental design is the most important section, hence, well-water and water-deficit must be narrated in details.

We have provided such details. Please see 158-159. Further details are in supplementary methods S1. 

7. Subsection 2.4.2 " Investigation of G x E interaction in the MET dataset", no cited reference was used, therefore, it should mention as the methodology developed by the authors.

We have now detailed (lines 22-226) and provided reference (Henderson, C. (1950). Estimation of Genetic Parameters. Annals of Mathematical Statistics 21, 309-310). Sorry we could not format this according to Plants endnote style, as it is plants already provided us copyedited version. we will do this once it is accepted.

8. The result sections were well written, however, Fig 5 should be replaced by another one with high quality and solution.

We have provided now the new figure 5. please see line 474-477

9. One of the weakest point of this study was to the experiments were done one only under rainout shelters, hence the attained results may not be reliable. Therefore, the discussion must be mentioned this points in detail.

This is already discussed. Please see lines 626-631

10. All cited references must be listed in number following the style of Plants journal

We have reformatted according to the plants style using Endnotes. However, some of the references could be changed in the version that Plants provided us. we will work with the journal to ensure this is done accurately. Best Regards and thanks for the comments

Round 2

Reviewer 1 Report

The authors corrected the text according to what was proposed in the review of the article. I have no more proposals for amendments.

Author Response

There is no comment to address for this review report.

Reviewer 4 Report

The revised manuscript is fine now. All the comments and suggestions have clearly explained and revised. However, All the references listed in the reference section must be edited following the style of Plants, for instance, the names of journals should be abbreviated.